Acute effects of variable resistance training on force, velocity, and power measures: a systematic review and meta-analysis

Shi Lin 1
Cai Zhidong 1
Chen Sitong 2
Han Dong handtiyu@126.com 1
1 School of Physical Education and Sport Training, Shanghai University of Sport , Shanghai , China
2 Institute for Health and Sport, Victoria University , Melbourne , Australia
Mendez-Rebolledo Guillermo
Electronic publication date: 2022 Aug 17
Publication date: 2022
Volume: 10
Electronic Location ID: e13870
Received 2022 Apr 14; Accepted 2022 Jul 19
Copyright: ©2022 Shi et al.
Copyright year: 2022
Copyright holder: Shi et al.
License: This is an open access article distributed under the terms of the Creative Commons Attribution License, which permits unrestricted use, distribution, reproduction and adaptation in any medium and for any purpose provided that it is properly attributed. For attribution, the original author(s), title, publication source (PeerJ) and either DOI or URL of the article must be cited.
License URL: https://creativecommons.org/licenses/by/4.0/

Keywords: Elastic bands, Chains, Neuromuscular, Strength training

Funding: The authors received no funding for this work.

==============================
Objective

Acute effects of variable resistance training (VRT) and constant resistance training (CRT) on neuromuscular performance are still equivocal. We aimed to determine the differences between VRT and CRT in terms of force, velocity, and power outcomes.

Methods

We searched PubMed, Web of Science, and SPORTDiscus electronic databases for articles until June 2021. Crossover design studies comparing force, velocity, and power outcomes while performing VRT and CRT were included. Two reviewers independently applied the modified version of the Cochrane Collaboration’s tool to assess the risk of bias. A three-level random effects meta-analyses and meta-regressions were used to compute standardized mean differences (SMDs) and 95% confidence intervals.

Results

We included 16 studies with 207 participants in the quantitative synthesis. Based on the pooled results, VRT generated greater mean velocity (SMD = 0.675; moderate Grading of Recommendations Assessment, Development and Evaluation (GRADE) quality evidence) and mean power (SMD = 1.022; low) than CRT. Subgroup analyses revealed that VRT considerably increased the mean velocity (SMD = 0.903; moderate) and mean power (SMD = 1.456; moderate) in the equated loading scheme and the mean velocity (SMD = 0.712; low) in the CRT higher loading scheme. However, VRT marginally significantly reduced peak velocity (SMD = −0.481; low) in the VRT higher loading scheme. Based on the meta-regression analysis, it was found that mean power (p = 0.014–0.043) was positively moderated by the contribution of variable resistance and peak velocity (p = 0.018) and peak power (p = 0.001–0.004) and RFD (p = 0.003) were positively moderated by variable resistance equipment, favoring elastic bands.

Conclusions

VRT provides practitioners with the means of emphasizing specific force, velocity, and power outcomes. Different strategies should be considered in context of an individual’s needs. Systematic review registration: PROSPERO CRD42021259205.

Introduction

Resistance training has been widely used to improve strength, speed, and power in athletes, which are the main determinants of most sports that involve jumping, sprinting, and change of direction (Suchomel et al., 2018; Suchomel, Nimphius & Stone, 2016). Traditional resistance training, which employs isoinertial training, is known as constant resistance training (CRT) (Frost, Cronin & Newton, 2010). Specifically, the load lifted by an individual is constant in the range of motion. However, training in this manner is not without constraints. For instance, in most multijoint exercises (e.g., back squat, bench press, and deadlift), muscle force production decreases in a disproportional manner in the early phase of concentric movement owing to mechanical disadvantages at specific joint angles (Elliott, Wilson & Kerr, 1989), which may result in the deceleration of upward movement (Van den Tillaar, Andersen & Saeterbakken, 2014; Van den Tillaar & Ettema, 2010). This common occurrence is pervasively called the “sticking point” in the context of resistance training (Kompf & Arandjelović, 2016; Kompf & Arandjelović, 2017). When movement extends beyond the sticking point, the larger internal and smaller external moment arms, which develop at the hip and knee joints (Elliott, Wilson & Kerr, 1989), ensure that muscle force production increases (i.e., a mechanical advantage), thereby resulting in an increased upward velocity (Martinez-Cava et al., 2019; Van den Tillaar & Ettema, 2010). Several multijoint exercises have such a strength curve that follows an ascending pattern throughout the concentric range of motion (Wallace, Bergstrom & Butterfield, 2018). Therefore, maximal muscle activation only occurs in the early phase of concentric movement in the context of CRT.

Various training methods have been developed to address the constraints of CRT. Variable resistance training (VRT) is a method that can accommodate some multijoint exercises with ascending strength curve (Wallace, Bergstrom & Butterfield, 2018). The defining characteristics of VRT include the provision of unloading where muscle force production is compromised and overloading where muscle force production is the greatest (Fleck & Kraemer, 2014). Anecdotal evidence of VRT dates back to the 1900s, when new training devices, such as cams, were developed in an attempt to combat the mechanical disadvantages associated with CRT (Frost, Cronin & Newton, 2010). However, some limitations that restrict the use of cams include the capital outlay on the device (Haff, 2000) and difficulty in combining cams with free weights (McMaster, Cronin & McGuigan, 2009). In contrast, the addition of elastic bands or chains to CRT as makeshift cams has recently gained attention owing to the relative portability and inexpensiveness of these equipment (Suchomel et al., 2018). Training in this manner, the loading an individual experiences is gradually increased in the concentric range of motion and vice versa.

Many studies have compared acute neuromuscular responses while performing VRT and CRT; however, the existing evidence is somewhat conflicting. Several studies showed that VRT was superior to CRT in terms of improving force outcomes (Andersen et al., 2020; Israetel et al., 2010; Kubo et al., 2018; Swinton et al., 2011; Wallace, Winchester & McGuigan, 2006), whereas others did not (Galpin et al., 2015; Nijem et al., 2016); moreover, other studies found no difference between the two training strategies (Coker, Berning & Briggs, 2006; Ebben & Jensen, 2002). Similar to the force outcome results, evidence supporting improved velocity and power while performing VRT is also disputed (Galpin et al., 2015; Swinton et al., 2011). In particular, these inconsistent results can be attributed to the different VRT design methodologies used, including the method of equating the loading schemes (e.g., whether the relative loading is equated between VRT and CRT), the contribution of variable resistance (i.e., how much of the loading is coming from the elastic bands or chains), and variable resistance equipment differences (the training stimuli is governed by the inertial properties of these two equipment (Arandjelovic, 2010; Frost, Cronin & Newton, 2010)). For example, VRT using a relatively higher loading scheme (i.e., the loading at the bottom position is equal between VRT and CRT) significantly decreased peak velocity compared to CRT (Saeterbakken, Andersen & Van den Tillaar, 2016; Stevenson et al., 2010), some studies found that VRT using a relatively lower loading scheme (i.e., the loading at the top position is equal) increased peak and mean velocity compared to CRT (Baker & Newton, 2009; Heelas, Theis & Hughes, 2021). Although these results seem to be plausible, there were also some discrepancies in studies using same VRT design methodology. For example, when using equated loading scheme (i.e., the loading is lower at the bottom and higher at the top in VRT than CRT), several studies demonstrated an increased force while performing VRT compared to CRT (Andersen et al., 2020; Kubo et al., 2018), one study found a decreased mean and peak force while performing VRT compared to CRT (Galpin et al., 2015). Thus, current studies are discrepant about the validity of VRT and whether acute neuromuscular responses (e.g., force and velocity) are actively affected while performing VRT.

VRT influences the magnitude of acute neuromuscular responses, and consequently long-term training adaptations. Nilo dos Santos et al. (2018) determined that there was no statistically significant difference between VRT and CRT in a training intervention meta-analysis in which maximum strength performance, which was previously found to improve following VRT compared with CRT, was investigated (Soria-Gila et al., 2015). This result was likely caused by a lack of consistency in the VRT methodologies used across studies, i.e., using an inappropriate VRT design method may produce adverse neuromuscular adaptations, leading to temporal attenuation of the training effect. Therefore, it is necessary to investigate the mechanics underlying the effects of VRT, which will provide pertinent information to help practically apply VRT strategies and further optimize training adaptations.

To the best of our knowledge, no systematic reviews have investigated the difference in acute neuromuscular responses between VRT and CRT. Therefore, the objective of the review was to collate evidence from crossover studies to (1) compare the acute neuromuscular responses (i.e., force, velocity, and power variables) while performing VRT and CRT; (2) investigate potential differences on the loading schemes, the contribution of variable resistance, and variable resistance equipment. The results of this meta-analysis will be useful for strength and conditioning practitioners to better understand the effectiveness of VRT and the specificity of different VRT methodologies, and may enable better prescribe VRT protocols.

Methods

This systematic review and meta-analysis was conducted in accordance with the Preferred Reporting Items for Systematic Reviews and Meta-Analyses (PRISMA) 2020 statement (Page et al., 2021). The protocol was registered with the International Prospective Register of Systematic Reviews (PROSPERO; Registration number: CRD42021259205).

Search strategy

The electronic databases of PubMed, Web of Science, and SPORTDiscus were used for performing a search of articles from their inception to June 3, 2021. The following search strategy was adapted for each database and combined under Boolean’s language: “elastic band” OR “rubber band” OR “thera-band” OR “elastic tubing” OR “chain” AND “variable resistance” OR “accommodating resistance” OR “resistance training” OR “free weight” OR “back squat” OR “bench press” OR “deadlift” OR “weightlifting” AND “kinetic” OR “kinematic” OR “force” OR “power” OR “velocity”. The detailed search strategy for each database is shown in the Supplementary Material (Table S1). After removing duplicates, the title and abstract of each article were screened for potential inclusion. The full-text studies were excluded, along with the reasons for exclusion. The reference lists in the selected studies were screened for additional related studies. Conference abstracts and proceedings were excluded.

Eligibility criteria

In accordance with the PICOS model (Page et al., 2021), studies were selected if they satisfied the following criteria:(1) population: healthy adults of both sexes; (2) intervention: an exercise group that adding elastic bands or chains to barbell weight training; (3) comparator: an exercise group that used barbell weight training; (4) outcomes: one or more of peak or mean force, velocity, and power variables directly captured while performing VRT and CRT using force plate, linear transducer or motion capture equipment; (5) study design: acute and crossover design; (6) the study was peer-reviewed and published in English. Studies were excluded if any of the following criteria were met: (1) the VRT group did not use a barbell weight; (2) the VRT group used elastic bands that were fixed in the upper position (deloading set-up); (3) outcomes were measured in an athletic performance test or only electromyography was reported; (4) data (mean  ± standard deviation) only reported in graphical form that were not extracted from WebPlotDigitizer software (version 4.5) and the authors could not be contacted.

Study selection and data extraction

The first author imported all records into Endnote X9 software (Clarivate Analytics, Philadelphia, PA, USA) and deleted the duplicates. The first and second authors separately screened the study titles and abstracts before retrieving and assessing the full texts for eligibility. Disagreements in eligibility were resolved through consultation with the third author. Data extraction was performed by the first author using an Excel spreadsheet, checked by the second author, and any differences were resolved through discussion, or through consultation with the third author. The following data were extracted from the included studies: (1) publication details, including author and year; (2) participant characteristics, including sample size, sex, age, training experience and strength levels; (3) condition prescription details for VRT and CRT, including the number of sets, repetitions, loading and loading comparison; and (4) outcome measures, including peak and/or mean force, velocity, and power. Quantitative data (means and standard deviation) were extracted from the text. If insufficient information was reported, the authors of those studies were contacted by email to obtain missing information. When data was only reported in figures and the authors did not provide the requested data, WebPlotDigitizer software (version 4.5) was used for data extraction. In case where standard errors were reported, they were converted to standard deviation post hoc.

Dependent variables

Outcomes of interest in the review included force (peak and/or mean force), velocity (peak and/or mean velocity), and power (peak and/or mean power and rate of force development (RFD)).

Risk of bias assessment and certainty of evidence

A modified version of the Cochrane Collaboration’s tool for assessing risk of bias was used (Higgins et al., 2011). Modifications included adding familiarization, measurement, effort, and intensity bias criteria and removing the performance bias. The performance bias was removed because the blinding of participants and researchers was considered impossible (Jukic et al., 2020). Familiarization bias was related to whether the participants were adequately familiarized with VRT. Measurement bias was related to whether the included studies used appropriate instruments to assess the variable resistance. For instance, a study may have induced bias if the variable resistance was estimated using an equation or manufacturer’s table. Effort bias was related to whether the authors clearly reported that the participants were encouraged to perform the lift as fast as possible. If there was no such statement, force and velocity might be affected by different tempos of lifting. Intensity bias was a quantitative assessment of whether the loading matched between VRT and CRT. If the quantitative assessment of load was not equal between VRT and CRT, bias may have been induced. The risk of bias assessments was performed independently by two authors, and disagreements were resolved through discussion.

The certainty of evidence in each meta-analysis was rated using the Grading of Recommendations Assessment, Development and Evaluation (GRADE) system (Atkins et al., 2004), similarly to previous meta-analysis reviews comparing biomechanical outcomes between different training modalities (Jukic et al., 2020; Miller et al., 2019; Van Hooren et al., 2020). To summarize, the quality was rated as high and then downgraded one level to moderate, low or very low for each outcome based on the following domains: (1) total sample size < 100 participants (imprecision), (2) statistically significant heterogeneity (inconsistency), and (3) more than 50% of studies in a meta-analysis had > 1 risk of bias item assessed to be high risk (risk of bias).

Statistical analysis

Studies reported on one or multiple outcomes using different contributions of variable resistance or variable resistance equipment. Therefore, for most studies, multiple effect sizes were included. An important requirement in traditional meta-analytic approaches is that there is independency of effect sizes in the dataset, suggesting that only one effect size should be included per study, so multiple effect sizes extracted from the same study inflated the traditional meta-analysis results (Assink et al., 2015). To address the dependency, the present study applied a three-level random effects model to analyze the data (Assink & Wibbelink, 2016). A three-level random effects model accounts for three sources of variance: sampling variance (level 1), the variance between effect sizes from the same study (level 2), and the variance between studies (level 3) (Van den Noortgate et al., 2013). The heterogeneity within (level 2) and between (level 3) studies can be assessed accordingly. The analyses were performed in R (version 4.2.1-win), using the function “rma.mv” of the metafor package (Viechtbauer, 2010).

The standardized mean difference (SMD) with Cohen’s d and 95% confidence intervals were calculated between CRT and VRT based on the means and standard deviations of force, velocity, and power outcomes. The SMD magnitude was interpreted as small (0.20–0.49), moderate (0.50–0.79), or large (>0.80) (Cohen, 1988). In addition to the overall analysis (including all loading schemes), we performed separate meta-analyses for the three loading schemes: the equated loading scheme (the loading is lower at the bottom and higher at the top in VRT than in CRT), the CRT higher scheme (the loading at the top is equal to VRT and lower at the bottom), and the VRT higher scheme (the loading at the bottom is equal to CRT and higher at the top). Meta-regressions were conducted when at least six effects were available for a certain outcome (Fu et al., 2011). The equipment (chains or elastic bands) was the categorical covariate, and the contributions of variable resistance in VRT were continuous covariates. Furthermore, three studies (Andersen et al., 2020; Ebben & Jensen, 2002; Saeterbakken, Andersen & Van den Tillaar, 2016) in which loading was reported as a repetition maximum were converted to percentages of the one-repetition maximum (1-RM) based on the National Strength and Conditioning Association standard (Haff & Triplett, 2016) to normalize the analyses. Egger’s test was used to assess for publication bias. A p-value of <0.05 was considered statistically significant.

Results

Search results

A flow diagram of the search and screening process is presented in Fig. 1. The initial search retrieved 646 articles, and 422 articles remained after removing the duplications. The title and abstract screening excluded an additional 398 articles, and 24 full-text articles were assessed for eligibility. Based on the inclusion criteria, 11 articles were rejected. Six additional studies were discovered via screening of citations. Two studies (Andersen et al., 2016; Saeterbakken et al., 2014) only reported electromyography outcomes, and data from one study (Israetel et al., 2010) were not available. As a result, 16 studies were included for quantitative synthesis in this review.

Figure 1 Literature search flow diagram.

n, number of studies; CRT, constant resistance training; VRT, variable resistance training.

Study characteristics

Detailed study characteristics are reported in Table 1. The total number of participants was 207 (165 males and 42 females). Of the 16 included studies, two studies included only females, four studies included both males and females, and the remaining studies included only men. Ten studies recruited participants who had at least one year of training experience, participants in three studies had at least six months of training experience, and the amount of training experience of participants was unclear in three studies. Participants performed back squat in five studies, bench press in two studies, bench throw in one study, deadlift in five studies, and Olympic lifts in three studies. Ten studies used elastic bands as the variable resistance equipment, and seven studies used chains as the variable resistance equipment. The mean contribution of variable resistance in the included studies was 36% 1-RM (range: 5%–48% 1-RM), and the mean free weight ranged from 10% to 90% 1-RM (median = 50% 1-RM). Six studies utilized an equated loading scheme between conditions, seven studies utilized a CRT higher scheme, and three studies utilized a VRT higher scheme.

Table 1 Summary of the studies pertinent to kinematics and kinetics to VRT and CRT.

Study	M/F; age (mean ± SD)	Training experience; relative strength levels (BM/1-RM)	Exercise; Equipment	Sets × repetitions	CRT loading	VRT loading (Free weighs + VRT (bottom–top))	Loading comparison	Outcomes	
Andersen et al. (2020)	16/0; 23 ± 2 y	3 years; 2-RM:1.8	Deadlift; Elastic bands	1 × 1	2-RM (95% 1-RM)	2-RM (74% 1-RM + 0–48% 1-RM)	Equal	PF, MF, PV, MV	
Baker & Newton (2009)	13/0; 20 ± 3 y	Unclear; 1.3	Bench press; Chains	2 × 3	75% 1-RM	60% 1-RM + 0–14% 1-RM	CRT ↑	Concentric/ eccentric PV, concentric MV	
Berning, Coker & Briggs (2008)	4/3; 31 ± 12 y	Trained: 6 years; 0.8	Clean; chains	1 × 1	a: 80% 1-RM b: 85% 1-RM	a: 75% 1-RM + 5% 1-RM b: 80% 1-RM + 5% 1-RM	CRT ↑	PF, PV, RFD	
Coker, Berning & Briggs (2006)	4/3; 31 ± 12 y	Trained: 6 years; 0.8	Snatch; chains	1 × 1	a: 80% 1-RM b: 85% 1-RM	a: 75% 1-RM + 5% 1-RM b: 80% 1-RM + 5% 1-RM	CRT ↑	PF, PV, RFD	
Ebben & Jensen (2002)	5/6; 19 ± 2 y	Trained: unclear; unclear	Back squat; a: Elastic bands, b:Chains	1 × 5	5-RM	a: 5-RM (9% 1-RM) b: 5-RM (9% 1-RM)	CRT ↑	PF, MF	
Galpin et al. (2015)	12/0; 24 ± 2 y	Trained: at least 6 months; 2.2	Deadlift; Elastic bands	1 × 3	a: 60% 1-RM b: 85% 1-RM	a1: 55% 1-RM + 0–9% 1-RM a2: 50% 1-RM + 0–21% 1-RM b1: 78% 1-RM + 0–13% 1-RM b2: 70% 1-RM + 0–30% 1-RM	Equal	PF, MP, PV, MV, PP, MP, RFD	
Garcia-Lopez et al. (2016)	a: 8 rugby players, b: 8 undergraduate students/0; a: 27 ± 4 y. b: 21 ± 1 y	Mixed; a: 1.4, b: 1.1	Bench press; Elastic bands	A set to failure	85% 1-RM	67% 1-RM + mean 18% 1-RM	Equal	PV, MV	
Godwin, Fernandes & Twist (2018)	8/0; 19 ± 2 y	Trained: at least 2 years; 1.2	Bench press throw; Chains	1 × 3	45% 1-RM	30% 1-RM + unclear-15% 1-RM	CRT ↑	PV, MV, PP, MP	
Heelas, Theis & Hughes (2021)	15/0; 29 ± 9 y	Trained: at least 1 year; 2.1	Deadlift; Elastic bands	1 × 6	54% 1-RM	a: 43% 1-RM + 0–11% 1-RM b: 41% 1-RM + 0–13% 1-RM c: 38% 1-RM + 0–16% 1-RM	CRT ↑	PV, MV, PP, MP	
Kampanart, Chaninchai & Chaipat (2016)	0/6; 17 ± 2 y	Trained: at least 4 years; Unclear	Clean pull; Elastic bands	3 × 3	90% 1-RM	a: 90% 1-RM + 0–9% 1-RM b: 90% 1-RM + 0–18% 1-RM	VRT ↑	PF, PV, PP	
Kubo et al. (2018)	10/0; 23 ± 2 y	At least 1 year; 1.6	Back squat; Elastic bands	1 × 3	56% 1-RM	a: 45% 1-RM + 11% 1-RM b: 34% 1-RM + 22% 1-RM c: 22% 1-RM + 34% 1-RM d: 11% 1-RM + 45% 1-RM	Equal	MF, MV, MP	
Nijem et al. (2016)	13/0; 24 ± 2 y	At least 6 months; 2	Deadlift; Chains	1 × 3	85% 1-RM	68% 1-RM + 17% 1-RM	CRT ↑	PF	
Saeterbakken, Andersen & Van den Tillaar (2016)	0/20; 23 ± 3 y	Trained: at least 5 years; 6-RM:1.1	Back squat; Elastic bands	1 × 6	6-RM (85% 1-RM)	6-RM (83%–95% 1-RM)	VRT ↑	PV	
Stevenson et al. (2010)	20/0; 26 ± 4 y	Trained: at least 10 years; Unclear	Back squat; Elastic bands	3 × 3	55% 1-RM	55% 1-RM + 0–11% 1-RM	VRT ↑	concentric/ eccentric PV, concentric/ eccentric MV, PP, RFD	
Swinton et al. (2011)	23/0; 27 ± 6 y	Trained: at least 10 years; 2.1	Deadlift; Chains	1 × 2	a: 30% 1-RM b: 50% 1-RM c: 70% 1-RM	a1: 20% 1-RM + 0–20% 1-RM a2: 10% 1-RM + 0–40% 1-RM b1: 40% 1-RM + 0–20% 1-RM b2: 30% 1-RM + 0–40% 1-RM c1: 60% 1-RM + 0–20% 1-RM c2: 50% 1-RM + 0–40% 1-RM	Equal	PF, PV, PP	
Wallace, Winchester & McGuigan (2006)	6/4; 21 ± 2 y	At least 6 months; Unclear	Back squat; Elastic bands	2 × 3	a: 60% 1-RM b: 85% 1-RM	a1: 54% 1-RM + 0–12% 1-RM a2: 50% 1-RM + 0–21% 1-RM b1: 74% 1-RM + 0–17% 1-RM b2: 70% 1-RM + 0–30% 1-RM	Equal	PF, PP, RFD	
Notes.

M male

F female

CRT constant resistance training

VRT variable resistance training

BM body mass

RM repetition maximum

PF peak force

MF mean force

PV peak velocity

MV mean velocity

PP peak power

MP mean power

RFD rate of force development

↑ high

Risk of bias

The results of modified risk of bias assessment are summarized in the Supplementary Material (Fig. S1). Five studies were at a high risk of order effect, either reporting a fixed starting condition or not reporting information about randomization. Only two studies were classified as low risk. None of the studies reported information on allocation concealment or blinding outcome assessment. Seven studies did not refer to a familiarization session. Two studies were at a high risk of measurement error owing to the use of a regression equation to estimate variable resistance. Three studies were at a high risk of effort bias because the participants were not encouraged to perform the lift as fast as possible. Seven studies matched the loading between conditions.

Meta-analysis

Force outcomes

The pooled effect size of the peak force (SMD = 0.051 (−0.258, 0.36); p = 0.737; k = 25) and mean force (SMD = 0.207 (−0.873, 1.287); p = 0.678; k = 11) were not found to be significantly different between conditions. Next, it was also found that VRT did not significantly affect peak force and mean force in each loading scheme in comparison with CRT (Table 2). All subgroup and pooled results provided low GRADE quality of evidence. The variable resistance equipment and the contribution of variable resistance were not significant moderators for both peak force and mean force (Table 3). No evidence of publication bias was observed for peak force (Egger’s test: t = −1.00, p = 0.326) and mean force (Egger’s test: t = −0.82, p = 0.445).

Table 2 Results for overall effect sizes of specific scheme and quality of evidence assessment.

Outcome	Summary of findings	Quality evidence assessment (GRADE)	
	k	n	Pooled effect (95% CI)	p-value	Variance level 2a	Variance level 3b	Imprecision	Inconsistency	Risk of bias	Overall quality	
Peak force (N)											
Equated loading scheme	15	61	0.193 (−0.532, 0.918)	0.577	0.000	0.399*	−1	−1	None	Low	
VRT higher scheme	3	26	0.081 (−0.992, 1.155)	0.775	0.000	0.000	−1	None	−1c	Low	
CRT higher scheme	7	38	−0.112 (−0.55,0.326)	0.555	0.000	0.000	−1	None	−1c	Low	
All	25	125	0.051 (−0.258,0.36)	0.737	0.000	0.131*	None	−1	−1c	Low	
Mean force (N)											
Equated loading scheme	9	38	0.308 (−0.54, 0.9)	0.661	0.048	1.273*	−1	−1	None	Low	
CRT higher scheme	2	11	−0.085 (−3.897, 3.727)	0.824	0.000	0.000	−1	None	−1c	Low	
All	11	49	0.207 (−0.873, 1.287)	0.678	0.000	0.859*	−1	−1	None	Low	
Peak velocity (m/s)											
Equated loading scheme	13	67	0.041 (−1.628, 1.71)	0.958	0.000	2.261**	−1	−1	None	Low	
VRT higher scheme	4	46	−0.481 (−1.109, 0.148)	0.093	0.000	0.000	−1	None	−1c	Low	
CRT higher scheme	9	50	0.361 (−0.238, 0.96)	0.202	0.000	0.198	−1	None	−1c	Low	
All	26	163	0.024 (−0.552, 0.601)	0.931	0.000	0.819**	None	−1	−1c	Low	
Mean velocity (m/s)											
Equated loading scheme	11	54	0.903 (0.303, 1.504)	0.007*	0.000	0.202	−1	None	None	Moderate	
VRT higher scheme	1	20	−0.27 (−0.90, 0.35)	0.387	Not applicable					
CRT higher scheme	5	36	0.712 (−0.216, 1.641)	0.1	0.000	0.213	−1	None	−1c	Low	
All	17	110	0.675 (0.206, 1.144)	0.008*	0.000	0.288*	None	−1	None	Moderate	
Eccentric peak velocity (m/s)											
VRT higher scheme	1	20	0.23 (−0.39, 0.86)	0.04*	Not applicable					
CRT higher scheme	1	13	0.85 (0.04, 1.65)	0.462	Not applicable					
All	2	33	0.484 (−3.39,4.357)	0.358	0.029	0.029	−1	None	−1c	Low	
Peak power (W)											
Equated loading scheme	14	45	−0.144 (−1.341, 1.053)	0.799	0.017	0.879**	−1	−1	None	Low	
VRT higher scheme	3	26	0.067 (−1.01, 1.144)	0.814	0.000	0.000	−1	None	−1c	Low	
CRT higher scheme	4	23	0.46 (−0.188, 1.108)	0.109	0.000	0.003	−1	None	−1c	Low	
All	21	94	0.031 (−0.509, 0.571)	0.906	0.000	0.384**	−1	−1	−1c	Low	
Mean power (W)											
Equated loading scheme	8	22	1.456 (0.165, 2.748)	0.032*	0.119	0.505	−1	None	None	Moderate	
CRT higher scheme	4	23	0.615 (−0.589, 1.819)	0.203	0.000	0.169	−1	None	−1c	Low	
All	12	45	1.022 (0.241, 1.804)	0.015*	0.000	0.410*	−1	−1	None	Low	
RFD (N)											
Equated loading scheme	14	45	0.079 (−0.42, 0.578)	0.738	0.000	0.125*	−1	−1	None	Low	
VRT higher scheme	1	20	0.24 (−0.39. 0.86)	0.457	Not applicable					
CRT higher scheme	5	27	−0.39 (−1.01, 0.23)	0.156	0.000	0.000	−1	None	−1c	Low	
All	20	92	−0.043 (−0.347, 0.261)	0.77	0.000	0.073*	−1	−1	None	Low	
Notes.

CI confidence interval

CRT constant resistance training

VRT variable resistance training

RFD rate of force development

GRADE Grading of Recommendations Assessment, Development and Evaluation

k number of effect sizes

n number of participants

a Variance between the effect sizes extracted from the same study.

b Variance between studies.

c More than 50% of studies had > 1 risk of bias item assessed to be high risk.

* p < 0.05.

** p < 0.001.

Table 3 Results for categorical and continuous moderators.

Moderator variables	k	Intercept/mean d (95% CI)	β (95% CI)	F (df1, df2)	p	Variance level 2a	Variance level 3b	
(1) Peak force								
Equated loading scheme								
Equipment				F(1, 13) = 0.038	0.848	0.000	0.651*	
Elastic bands	9	0.156 (−0.908, 1.219)						
Chains	6	0.342 (−1.422, 2.105)	0.186 (−1.874, 2.246)					
Contribution of VR	15	−0.122 (−0.963, 0.72)	1.081 (−1.025, 3.186)	F(1, 13) = 1.23	0.288	0.000	0.256*	
CRT higher scheme								
Equipment				F(1, 5) = 0.065	0.809	0.000	0.000	
Elastic bands	1	−0.21 (−1.301, 0.881)						
Chains	6	−0.091 (−0.599, 0.417)	0.119 (−1.084, 1.322)					
Contribution of VR	7	0.373 (−0.649, 1.395)	−5.493 (−15.818, 4.833)	F(1, 5) = 1.87	0.23	0.000	0.000	
All								
Equipment				F(1, 23) = 0.006	0.938	0.000	0.151*	
Elastic bands	13	0.041 (−0.378,0.46)						
Chains	12	0.064 (−00402, 0.53)	0.023 (−0.579, 0.625)					
Contribution of VR	25	−0.195 (−0.598, 0.208)	1.304 (−0.286, 2.894)	F(1, 23) = 2.877	0.103	0.000	0.082*	
(2) Mean force								
Equated loading scheme								
Contribution of VR	9	−0.241 (−2.241, 1.761)	1.824 (−2.431, 6.078)	F(1, 7) = 1.027	0.345	0.298	1.072*	
All								
Equipment				F(1, 9) = 0.019	0.894	0.036	0.885*	
Elastic bands	10	0.218 (−0.917, 1.354)						
Chains	1	0.132 (−1.55, 1.813)	−0.087 (−1.517, 1.343)					
Contribution of VR	11	−0.225 (−1.567, 1.117)	1.743 (−1.677, 5.163)	F(1, 9) = 1.329	0.279	0.022	0.757*	
(3) Peak velocity								
Equated loading scheme								
Equipment				F(1, 11) = 7.671	0.018*	0.000	0.654*	
Elastic bands	7	0.735 (−0.373, 1.844)						
Chains	6	−1.934 (−3.742, −0.126)*	−2.669 (−4.79, −0.548)*					
Contribution of VR	13	0.445 (−1.457, 2.347)	−1.24 (−4.046, 1.567)	F(1, 11) = 0.945	0.352	0.000	2.209**	
CRT higher scheme								
Equipment				F(1, 7) = 3.353	0.11	0.000	0.071	
Elastic bands	3	0.923 (0.105, 1.741)*						
Chains	6	0.146 (−0.436, 0.728)	−0.777 (−1.781, 0.226)					
Contribution of VR	9	−0.331 (−1.45, 0.789)	7.081 (−2.574, 16.736)	F(1, 7) = 3.008	0.126	0.000	0.052	
All								
Equipment				F(1, 24) = 1.107	0.303	0.000	0.789**	
Elastic bands	14	0.264 (−0.474, 1.002)						
Chains	12	−0.325 (−1.215, 0.565)	−0.589 (−1.745, 0.567)					
Contribution of VR	26	0.251 (−0.468, 0.971)	−1.236 (−3.534, 1.061)	F(1, 24) = 1.233	0.278	0.000	0.838**	
(4) Mean velocity								
Equated loading scheme								
Contribution of VR	11	0.139 (−1.29, 1.567)	2.316 (−1.022, 5.653)	F(1, 9) = 2.464	0.151	0.000	0.639	
All								
Equipment				F(1, 15) = 0.464	0.506	0.000	0.31	
Elastic bands	15	0.753 (0.207, 1.3)*						
Chains	2	0.375 (−0.674, 1.425)	−0.378 (−1.561, 0.805)					
Contribution of VR		0.196 (−0.612, 1.004)	2.079 (−0.603, 4.761)	F(1, 15) = 2.729	0.119	0.000	0.402*	
(5) Peak power								
Equated loading scheme								
Equipment				F(1, 12) = 12.676	0.004*	0.017	0.099	
Elastic bands	8	0.397 (−0.2, 0.994)						
Chains	6	−1.185 (−1.947, −0.423)*	−1.582 (−2.55, −0.614)*					
Contribution of VR	14	0.152 (−1.110, 1.415)	−1.353 (−4.001, 1.294)	F(1, 12) = 1.24	0.287	0.006	0.754**	
All								
Equipment				F(1, 19) = 14.148	0.001*	0.000	0.082	
Elastic bands	14	0.339 (−0.028, 0.705)						
Chains	7	−0.903 (−1.489, −0.317)*	−1.242 (−1.933, −0.551)*					
Contribution of VR	21	0.323 (−0.324, 0.97)	−1.69 (−4.067, 0.686)	F(1, 19) = 2.216	0.153	0.000	0.31**	
(6) Mean power								
Equated loading scheme								
Contribution of VR	8	0.494 (−0.782, 1.769)	4.301 (0.197, 8.406)*	F(1, 6) = 6.575	0.043*	0.000	0.226	
All								
Equipment				F(1, 10) = 1.957	0.192	0.000	0.303	
Elastic bands	11	1.238 (0.446, 2.01)*						
Chains	1	0.09 (−1.567, 1.747)	−1.148 (−2.976, 0.68)					
Contribution of VR	12	0.208 (−0.609, 1.024)	4.751 (1.207, 8.294)*	F(1, 10) = 8.924	0.014*	0.000	0.126	
(7) RFD								
Equated loading scheme								
Equipment				F(1, 12) = 4.276	0.061	0.000	0.028	
Elastic bands	8	0.304 (−0.116, 0.723)						
Chains	6	−0.28 (−0.729, 0.169)	−0.584 (−1.198, 0.031)					
Contribution of VR	14	0.052 (−0.681, 0.785)	0.125 (−2.112, 2.362)	F(1, 12) = 0.015	0.905	0.000	0.136	
All								
Equipment				F(1, 18) = 11.865	0.003*	0.000	0.000	
Elastic bands	9	0.296 (0.007, 0.584)*						
Chains	11	−0.305 (−0.531, −0.08)*	−0.601 (−0.967, −0.234)*					
Contribution of VR	20	−0.104 (−0.565, 0.357)	0.339 (−1.564, 2.241)	F(1, 18) = 0.14	0.713	0.000	0.084	
Notes.

CI confidence interval

CRT constant resistance training

VRT variable resistance training

VR variable resistance

RFD rate of force development

k number of effect sizes

a Variance between the effect sizes extracted from the same study.

b Variance between studies.

* p < 0.05.

** p < 0.001.

Velocity outcomes

The pooled effect size of the peak velocity was not found to be significantly different between conditions (SMD = 0.024 (−0.552, 0.601); p = 0.931; k = 26). Next, it was found that VRT marginally significantly reduced peak velocity in the VRT higher scheme (SMD = −0.481 (−1.109, 0.148); p = 0.093; k = 4), not significantly affect peak velocity in the equated loading scheme (SMD = 0.041 (−1.628, 1.71); p = 0.958; k = 13) and CRT higher scheme (SMD = 0.361 (−0.238, 0.96); p = 0.202; k = 9) in comparison with CRT. All subgroup and pooled results provided low GRADE quality of evidence (Table 2). The variable resistance equipment (equated loading scheme: F(1, 11) = 7.671; p = 0.018) was a significant moderator, and VRT using elastic bands had a greater effect on peak velocity than VRT using chains. No evidence of publication bias was observed (Egger’s test: t = −0.16, p = 0.871).

The pooled analyses showed that VRT significantly improved mean velocity compared with CRT (SMD = 0.675 (0.206, 1.144); p = 0.008; k = 17). Next, it was found that VRT significantly improved mean velocity in the equated loading scheme (SMD = 0.903 (0.303, 1.504); p = 0.007; k = 11) and marginally significantly improved mean velocity in the CRT higher scheme (SMD = 0.712 (−0.216, 1.641); p = 0.1; k = 5) but not in the VRT higher scheme (SMD = −0.27 (−0.90, 0.35); p = 0.387; k = 1) in comparison with CRT. Subgroup results for equated loading scheme and CRT higher scheme provided moderate and low respectively, while pooled results provided moderate GRADE quality of evidence. Neither the variable resistance equipment nor the contribution of variable resistance were significant moderators (Table 3). No evidence of publication bias was observed (Egger’s test: t = 1.44, p = 0.170).

The pooled effect size of the eccentric peak velocity was not found to be significantly different between conditions (SMD = 0.484 (−3.39, 4.357); p = 0.358; k = 2). Next, it was found that VRT allowed for significantly higher peak eccentric velocity in the CRT higher loading scheme (SMD = 0.85 (0.04, 1.65); p = 0.04; k = 1) but nor in the VRT higher loading scheme (SMD = 0.23 (−0.39, 0.86); p = 0.462; k = 1) in comparison with CRT. Pooled results provided low GRADE quality of evidence.

Power outcomes

The pooled effect size of the peak power was not found to be significantly different between conditions (SMD = 0.031 (−0.509, 0.571); p = 0.906; k = 21). Next, it was also found that VRT did not significantly affect peak power in each loading scheme in comparison with CRT (Table 2). All subgroup and pooled results provided low GRADE quality of evidence. The variable resistance equipment (pooled results: F(1, 19) = 14.148; p = 0.001; equated loading scheme: F(1, 19) = 12.676; p = 0.004) was a significant moderator, with the use of elastic bands resulting in greater effects of VRT. Egger’s test showed a significant publication bias (t = 2.77, p = 0.012).

The pooled analyses showed that VRT significantly improved mean power compared with CRT (SMD = 1.022 (0.241, 1.804); p = 0.015; k = 12). Next, it was found that VRT could significantly improve mean power in the equated loading scheme (SMD = 1.456 (0.165, 2.748); p = 0.032; k = 8) but not in the CRT higher scheme (SMD = 0.615 (−0.589, 1.819); p = 0.203; k = 4) in comparison with CRT. Subgroup results for equated loading scheme provided moderate while subgroup results for CRT higher scheme and overall results provided low GRADE quality of evidence. The contributions of variable resistance (pooled results: F (1, 10) = 8.924; p = 0.014; equated loading scheme: F(1, 6) = 6.575; p = 0.043) was s significant moderator, and using higher contribution of variable resistance had a greater effect on mean power. No evidence of publication bias was observed (Egger’s test: t = −0.55, p = 0.597).

The pooled effect size of the RFD (SMD = −0.043 (−0.347, 0.261); p = 0.77; k = 20) was not found to be significantly different between conditions. Next, it was also found that VRT did not significantly affect RFD in each loading scheme in comparison with CRT (Table 2). All sub-group and overall results provided low GRADE quality of evidence. The variable resistance equipment (pooled results: F(1, 18) = 11.865; p = 0.003) was a significant moderator, with the use of elastic bands resulting in greater effects of VRT. No evidence of publication bias was observed (Egger’s test: t = 0.18, p = 0.856).

Discussion

To the best of our knowledge, this is the first meta-analytical investigation that compared acute neuromuscular responses (i.e., force, velocity, and power) while performing VRT and CRT. The results suggest that velocity and power benefit from the use of VRT. In addition, the neuromuscular responses to VRT differ across different loading schemes, variable resistance equipment, and contributions of variable resistance, which are important considerations for prescribing a VRT strategy.

Force outcomes

Low GRADE quality evidence indicated no differences in the effects of VRT and CRT on peak and mean force. Similarly, low GRADE quality evidence indicated that no significant differences were found between the training strategies when the different loading schemes were compared separately. Notably, some studies only reported peak variables, whereas other studies only reported mean variables, which may have affected the results. Nevertheless, the positive effects of using VRT over CRT in the equated loading scheme (SMD: 0.193 to 0.308) were much larger than those observed in the CRT higher scheme (SMD: −0.085 to −0.112). In particular, the main difference in VRT in which the abovementioned loading schemes are used is the higher loading in the end range of motion when using the equated loading scheme rather than the CRT higher scheme. Biomechanically, the external (i.e., loading) moment arm decreases at the specific joints as the extremity extends in most multijoint exercises, and the musculoskeletal lever system gradually gains a mechanical advantage; therefore, the highest force can be exerted in leg or arm extensions (McMaster, Cronin & McGuigan, 2009). In this case, applying additional external loading in the end range of motion probably increases the number of stimulated motor units and the frequency of firing coming mostly from the motoneurons, which innervate fast muscle fibers. This speculation was partly evidenced by Andersen et al. (2016), who demonstrated that the quadriceps were activated to a great extent in the middle and upper position when performing VRT compared with CRT. Therefore, using VRT in the equated loading scheme might be an appropriate means for improving force production.

Notably, two studies using an equated loading scheme and deadlifts as exercises reported conflicting peak force results. More specifically, Galpin et al. (2015) reported a significant decline in peak force with the increase in elastic bands tension, while Swinton et al. (2011) found an increased peak force as the chain loading increases. Inconsistent findings of these two studies may likely be explained by the variable resistance equipment used in VRT. Researchers investigated the differences between elastic bands and chains, and reported that two equipment could generate different loading features during the concentric range of motion due to different physical properties made (McMaster, Cronin & McGuigan, 2009), therefore may contribute to dissimilar mechanical effects (Frost, Cronin & Newton, 2010). This possibility is further supported by Arandjelovic (2010), who used a mathematical equation to show that greater peak force is required when chains are used. In addition, considering that the positive effects of VRT in eccentric phase can improve the concentric muscle force, the lack of eccentric movement (e.g., a deadlift) in VRT may not be an optimal strategy for developing a concentric force. Specifically, some previous studies (Frost, Cronin & Newton, 2010; Kuntz, Masi & Lorenz, 2014; Wallace, Bergstrom & Butterfield, 2018) proposed that a greater eccentric velocity may be achieved while performing VRT because the increased loading at the top range of motion pushes the individual downward, improving the stretch-shortening cycle and concentric force. Further, Baker & Newton (2009) also postulated that a within-repetition post-activation potentiation effect would be generated in eccentric movement while performing VRT, allowing for greater concentric performance. Overall, the variable resistance equipment used and the type of exercise performed in VRT may be potential factors that can modulate the force outcome; future research to elucidate this aspect is warranted.

Velocity outcomes

Based on the pooled results, VRT was shown to generate a greater increase in mean velocity (moderate GRADE quality evidence), but not in peak velocity (low GRADE quality evidence), in comparison with CRT. When grouped according to loading schemes, low to moderate GRADE quality evidence indicated that VRT was beneficial in optimizing mean velocity in the CRT higher scheme (SMD: 0.712) and the equated loading scheme (SMD: 0.69). In contrast, low GRADE quality evidence indicated that peak velocity was marginally significantly lower when VRT was employed in the VRT higher scheme (SMD: −0.481). These results suggest that a relatively lower loading scheme in VRT is conducive to eliciting a greater velocity and vice versa. Several studies also demonstrated that VRT using an equated loading scheme is also advantageous to improve velocity (Galpin et al., 2015; Garcia-Lopez et al., 2016; Kubo et al., 2018). In theory, the loading experienced by an individual around the sticking point is lower in VRT compared with that in CRT, which could accelerate the movement in the initial concentric range of motion. When the extremity is extended, an individual could produce more force against increased loading in a biomechanically advantage position and thereby maintain greater velocity throughout the range of motion. However, it is worth noting that, in the equated loading scheme, a significant difference between VRT and CRT was detected for mean velocity (p = 0.007), whereas no difference was observed for peak velocity (p = 0.958). The discrepant result for these two variables could be explained by the lack of peak or mean variables provided in some studies. For example, although the study by Swinton et al. (2011) reported that VRT has negative effects on the mean velocity, it was not included in the current meta-analysis because of insufficient data; Kubo et al. (2018) also did not report peak velocity data whereas the mean velocity is considerably increased while performing VRT compared to CRT. Thus, the result of meta-analysis should be interpreted with caution due to the insufficient data provided by the two studies (Kubo et al., 2018; Swinton et al., 2011). In addition, a significant heterogeneity (variance level 3 = 2.261, p < 0.001) in peak velocity was evident in the equated loading scheme. This could be explained by the fact that the study of Swinton et al. (2011) found a gradually decreased peak velocity as the chains loading increased, which is in contrast with other studies that using elastic bands as variable resistance equipment (Galpin et al., 2015; Garcia-Lopez et al., 2016). Moreover, our meta-regression substantiates that the use of elastic bands was a significant moderator in the equated loading scheme (p = 0.018), favoring greater peak velocity. If this was the case, it would partly explain the discrepant findings in the abovementioned studies (Galpin et al., 2015; Garcia-Lopez et al., 2016; Swinton et al., 2011). Additionally, based on the evidence, it may be preferable to use elastic bands when the aim is to produce fast muscular contraction adaptations. Furthermore, it should also be emphasized that velocity may begin to plateau when a higher elastic band resistance is used, which may limit the power output (Heelas, Theis & Hughes, 2021; Wallace, Winchester & McGuigan, 2006). Thus, practitioners should understand that a threshold from the contribution of variable resistance may be reached before velocity is decreased.

Researchers proposed that VRT was conducive to increasing eccentric velocity, which would potentially result in a greater stretch-shortening cycle (McMaster, Cronin & McGuigan, 2009; Wallace, Bergstrom & Butterfield, 2018). We found low GRADE quality evidence that no significant difference was noted in eccentric velocity between VRT and CRT. However, each study (Baker & Newton, 2009; Stevenson et al., 2010) reported a significant and positive effect of VRT on peak eccentric velocity. The inclusion of only two studies in this meta-analysis may have lowered the statistical power and hence affected the pooled results. Based on limited evidence, it seems that VRT is more likely to improve eccentric velocity. Neurophysiologically, when the musculotendinous unit is rapidly stretched, elastic energy is stored and the muscle spindles are stimulated, which causes further reflexing muscle action (Bosco et al., 1982). Thus, both effects could increase force production in the concentric phase. Along with the highest force in the end range of motion potentially causing increased eccentric velocity, Baker & Newton (2009) also attributed the increased eccentric velocity to the lower loading at the bottom of the lift as the series elastic component became more compliant. This could explain the lack of differences in mean eccentric velocity observed using VRT in the VRT higher loading scheme (the loading is equal at the bottom between two training modalities) (Stevenson et al., 2010). Considering the above, using equated loading scheme might be more beneficial to produce fast eccentric velocity. Future research investigated this aspect is warranted.

Power outcomes

Power is defined as the product of force and velocity. Considering the similar force and greater velocity adaptations using VRT in the present meta-analysis, it is not surprising that significantly larger effects on mean power (low GRADE quality evidence; SMD: 1.022), but not peak power (low GRADE quality evidence; SMD: 0.031), were observed using VRT. Similar to the velocity outcomes, the inconsistent results between peak and mean variables on power could likely be explained by the data provided by different studies. For instance, only one study (Galpin et al., 2015) reported both mean and peak power variables; the other three studies only reported peak power (Swinton et al., 2011; Wallace, Winchester & McGuigan, 2006) or mean power (Kubo et al., 2018) respectively. We also did not include the mean power of the study of Swinton et al. (2011) due to insufficient data. Overall, only the study of Swinton et al. (2011) found that VRT had negative effects on power, and this was the only study in which chains were used as variable resistance equipment. Our meta-regression demonstrated that chains result in negative effects in VRT on peak power, similar to the results of the meta-regression on velocity. Based on these results, we can speculate that, although chains are considered advantageous for producing force (Arandjelovic, 2010; Frost, Cronin & Newton, 2010), the excessive decrease in velocity caused by chains may lead to reduced power. In addition, the meta-regression showed a greater positive effect of VRT on mean power with higher variable resistance. However, it should be noted that using larger variable resistance may attenuate power output. Specifically, two studies reported that power was hindered when the elastic band resistance increased to 16% 1-RM (Heelas, Theis & Hughes, 2021) and 30% 1-RM (Wallace, Winchester & McGuigan, 2006), respectively. Thus, as with the velocity outcome discussed above, the practitioner should be aware of the contribution of variable resistance to enhancing velocity and power. Moreover, it is worth mentioning that performing ballistic exercise in VRT may not be an optimal means to generate power adaptations (Godwin, Fernandes & Twist, 2018). A previous study indicated that peak velocity in a bench throw occurs near the release point (Newton et al., 1996). However, the loading at the end range of motion in VRT is the greatest, which could have a negative effect on peak velocity, ultimately attenuating the power output. Similarly, Kampanart, Chaninchai & Chaipat (2016) reported no difference in the peak power between clean pulls when using VRT with 18%1-RM variable resistance and CRT. Because of the limited number of studies using ballistic movements (Berning, Coker & Briggs, 2008; Coker, Berning & Briggs, 2006; Godwin, Fernandes & Twist, 2018; Kampanart, Chaninchai & Chaipat, 2016), future research in this area is required before a valid conclusion can be drawn.

RFD is another main parameter contributing to power output (Cormie, McGuigan & Newton, 2011). Our meta-analysis found no significant difference between VRT and CRT (low GRADE quality evidence; SMD: −0.043), which is surprising considering both velocity and power outcomes were benefited more from VRT. Upon further inspection, one study (Galpin et al., 2015) reported a significant increase in RFD when a higher elastic band loading (30% 1-RM) was used in a heavy load condition (85% 1-RM) but showed no difference when lifting was done in a moderate load condition (60% 1-RM), regardless of the contribution of variable resistance. Higher elastic band resistance may be more advantageous for synchronizing peak force, velocity, and power (Galpin et al., 2015). In theory, in the equated loading scheme, using higher elastic band resistance would cause a lower loading at the bottom in comparison with CRT, which is important for achieving greater velocity in the early concentric phase. Although velocity may be hindered as the variable resistance increases, the force would improve to a large extent in the biomechanically advantageous position. Thus, the interaction effect of force and velocity might ultimately increase the concentric RFD (Stevenson et al., 2010). In addition, compared to using chains as variable resistance equipment, the use of elastic bands resulting in greater positive effects of RFD (p = 0.003). Collectively, these results suggest that more loading in the biomechanically advantageous position is an important consideration for enhancing RFD.

Several limitations to this review should be acknowledged and addressed. First, some meta-analysis of pooled results and subgroups may have been affected by the lack of data, such as mean and peak velocity/power data, provided in specific studies; therefore, we encourage researchers to provide open data and analyze more variables (i.e., mean and peak force, velocity, and power) in biomechanical studies. Second, exercises performed in VRT may be a potential factor differentiating the effects of neuromuscular responses. Because the main intention of this review was to explore the differences in methodologies used in VRT, we did not classify exercises, and future studies should consider this factor. Third, several studies did not report the data required for this meta-analysis, and we therefore extracted this information from figures, which likely resulted in some error. Finally, muscle electromyography results were not included in our review because few studies investigated electromyography, and those that did, used different muscles and divided electromyography into different phases in the range of motion. Investigating muscle electromyography between VRT and CRT should be considered in future research.

This is the first systematic review to provide evidence by investigating acute neuromuscular differences between VRT and CRT. Based on the results obtained in this meta-analysis, we can provide suggestions for effectively prescribing VRT strategies with a specific training aim in practice as well as several directions for future research: (1) an equated loading scheme and higher contribution of variable resistance are recommended for VRT, but practitioners should note that the threshold for the contribution of variable resistance may be reached before a decline in velocity and power occurs; (2) elastic bands and chains may be more effective for enhancing velocity and force, respectively. However, studies that directly compare elastic bands with chains are lacking, and further research is required to distinguish the neuromuscular adaptations of these two equipment strategies; (3) non-stretch-shortening cycle and ballistic exercises might not be optimal for inducing selective mechanics, and research on the acute neuromuscular responses of different exercises performed in VRT is warranted; (4) anecdotal claims suggest that VRT induces different neuromuscular responses in the eccentric phase and the sticking point (McMaster, Cronin & McGuigan, 2009; Wallace, Bergstrom & Butterfield, 2018). However, these speculations cannot be confirmed because few studies investigated these aspects. Thus, we suggest that future research focuses more on these areas to elucidate the exact rationale behind VRT.

Conclusions

VRT was more effective in enhancing velocity and power. Specifically, VRT was superior to CRT in terms of improving velocity and power in both the equated loading and CRT higher schemes. Furthermore, utilizing elastic bands appeared to be more beneficial for increasing peak velocity and power. In addition, using equated loading scheme and higher contribution of variable resistance seems to be a better strategy to optimize neuromuscular adaptations, which may however attenuate the power output. Based on these findings, researchers and practitioners should take these factors into consideration to effectively implement VRT strategies in research and practice.

Supplemental Information

Supplemental Information 1 PRISMA checklist

Click here for additional data file.

Data S1 Raw data

Click here for additional data file.

Dataset S1 Dataset

Click here for additional data file.

Supplemental Information 4 Contribution & Rationle

Click here for additional data file.

Figure S1 Modified risk of bias assessment

Click here for additional data file.

Table S1 Search terms

Click here for additional data file.

Additional Information and Declarations

Competing Interests

Author Contributions

Data Availability

The authors declare there are no competing interests.

Lin Shi conceived and designed the experiments, performed the experiments, analyzed the data, prepared figures and/or tables, and approved the final draft.

Zhidong Cai conceived and designed the experiments, performed the experiments, analyzed the data, prepared figures and/or tables, and approved the final draft.

Sitong Chen conceived and designed the experiments, authored or reviewed drafts of the article, and approved the final draft.

Dong Han conceived and designed the experiments, authored or reviewed drafts of the article, and approved the final draft.

The following information was supplied regarding data availability:

The raw data is available in the Supplemental Files.

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
