# Peer review of "Acute effects of variable resistance training on force, velocity, and power measures: a systematic review and meta-analysis"

_PeerJ, doi:10.7717/peerj.13870_

## Round 0.1 · original submission · Major Revisions

From my point of view and that of the reviewers, the manuscript requires a number of Major Revisions. These are related to (i) deficit in the precise identification of the research problem, demonstrating the state of the art on the possible effects of variable resistance training compared with constant resistance training. Numerous evidence exists in this regard, even more reports on meta-analysis; ii) the systematic review presented lacks specificity in the inclusion and exclusion criteria of the articles; iii) although there is an assessment of the risk of bias, the current recommendations request an assessment of the certainty of the evidence, using instruments such as GRADE, understanding that the research included an intervention and a comparator; iv) the authors point out that the investigation and its report was based on PRISMA –statement that was updated in 2020– which is not fully applied in this investigation; v) the outcome(s) selected for the meta-analysis need to be clarified in more detail (without this information, the inclusion of variables that do not respond to the same construct may be misinterpreted), and vi) the Figures are not presented in a high-quality format (please follow PeerJ recommendations) which makes it difficult to review the values ​​of the Forest Plots statistics, especially those related to heterogeneity.

·

Basic reporting

This meta-analysis examined the effects of variable resistance training (VRT) in comparison to constant weight training (CRT) on the force, velocity, and power output in compound free weight exercises (i.e., squat, bench, and deadlift). The authors found that VRT effectively improves mean power and mean velocity compared to constant resistance training (CRT), particularly in the equated loaded scheme and CRT higher loading scheme. Also, VRT using elastic bands is more effective at improving power than weighted chains.

I felt the meta-analysis was thoroughly done, specifically using the PRSIM guidelines. The authors do an excellent job of stating the risk of assessment bias and used the proper statistical procedures to test for publication bias. The study extraction methods were correctly done, and the use of forest plots to show the SMD were good figures.

I have a few minor comments and one major comment.

Minor change: A meta-analysis was needed on this topic for the previous one in the Journal of Strength and Conditioning Research was retracted in 2015. Maybe the authors would like to mention this in the Introduction—just a suggestion. I have provided the links below.

Soria-Gila, MA, Chirosa, IJ, Bautista, IJ, Baena, S, and Chirosa, LJ. Effects of variable resistance training on maximal strength: A meta-analysis. J Strength Cond Res 29(11): 3260–3270, 201

Retraction: https://pubmed.ncbi.nlm.nih.gov/30540285/

Experimental design

Minor change: Page 12, under statistical analysis. I think the subgroup analyses should be explained a little better. For example, the CRT higher scheme states that the loading at the top is equal to VRT. Therefore, the loading at the bottom must be higher, which I believe should be in the definition. The same with the VRT higher scheme. It should be explained the loading at the bottom is equal to CRT and higher at the top—state where the loading is higher and lower at the bottom and the top positions. As a reviewer, I kept having to go back to this page to understand and remember the subgroup analysis definitions.

Validity of the findings

Major change: Methods: Did the author consider analyzing the quality of the studies selected using the Pedro scale? Typically, I see this done in most meta-analyses and presented in a table format.

Additional comments

Minor change: Page 20, under discussion. The authors state that VRT improves power and velocity, but its mean power and mean velocity. Neither peak value for each of those variables improved. The authors may want to clarify that.

Minor change: Page 22, Line 263. I would like to see another sentence or two explaining the benefit of an increased eccentric velocity. Yes, this does enhance a greater SSC, as they have stated. But more elaboration on the neurological or physiological benefits, with scientific references, would improve the discussion. Considering this is a significant factor in VRT elastic band training.

Minor change: Page 25, under discussion. I think another area of future research on this topic should be a meta-analysis on VRT but using training studies. The author does address this in their Introduction. Acute studies such as the ones presented here are good, however, long-term training effects I believe are more impactful. Training studies lasting > 6 weeks would be ideal.

·

Basic reporting

The article contains adequate scientific language and an order that makes it understandable and fluid to read. However, it lacks specificity in some topics that prevent justifying the problem and fully understanding the results.

Some observations are listed below:

a. The introduction is too long and causes the focus of the central theme to be lost. It is suggested to rethink and specify the background that justifies the review and meta-analysis carried out.

b. In the introduction there are very old references to justify the investigation. It is suggested to keep those that are exclusively necessary and update the rest (for example: Feigenbaum & Pollock, 1999; Abelbeck, 2002; Elliott et al., 1989; Escamilla et al., 2000, and others)

Experimental design

The design of the review and meta-analysis is appropriate, but several aspects need to be clarified.

a. It is declared that the research followed the recommendations of the Preferred Reporting Items for Systematic Reviews and Meta-Analyses statement guidelines. However, when reviewing the presentation of this manuscript, it is observed that they partially comply with these recommendations. In addition, the reference used corresponds to PRISMA 2015 and there is a 2020 update. It is suggested to use the most current (https://systematicreviewsjournal.biomedcentral.com/articles/10.1186/s13643-021-01626-4)

b. The inclusion criteria of the review of the articles are not indicated. This does not allow us to understand what were the criteria to exclude potentially eligible articles. It is suggested to specify.

c. It is not detailed what type of clinical trials were included in the review. That is, were all the studies randomized controlled clinical trials or did they also include non-randomized trials?. It is suggested to specify.

d. In the inclusion criteria it is indicated that the outcomes will be force, velocity, and power. Evaluated with any test or instrument? This could generate ambiguity in the conclusions in case it is a direct or indirect evaluation of the variable. It is suggested to specify.

e. Although a risk of bias assessment was applied, it could be complemented by an assessment of the methodological quality of the selected studies.

Validity of the findings

The research responds to an interesting research problem for the areas of physical activity, sports and health. The results are presented in an understandable way and the statistical analysis is appropriate for the proposed design. However, the results and conclusions could be ambiguous due to methodological deficiencies (previously indicated) of the systematic review.

a. The results do not describe the frequency and time of the interventions, which could be useful to enhance the study findings. This information is not observed in the summary table of the articles.

b. In the results, it is also not possible to understand how the outcomes related to the review were measured (instrument and/or test with which the study participants were evaluated). The outcomes were extracted from studies in which they evaluated with sprint, vertical jump, horizontal jump, isokinetic test, other...?

Additional comments

a. The title is promising. however, the concepts "kinematics" and "kinetics" seem too broad for the analyzed outcomes, which are very specific. It is suggested to modify the title so that it is more representative of the review and meta-analysis.

b. It would be interesting to read in the manuscript the physiological explanation why the VRT has a better response for velocity and power gain. At the muscle level, how different is the stimulation with VRT compared to CRT?

Reviewer 3 ·

Basic reporting

Thank you for the opportunity to review the manuscript. This meta-analysis aimed at comparing constant resistance training and variable resistance training on several acute measures such as power, velocity, and force. While this paper is interesting several points need to be considered before it is ready for publication.



Introduction
Lines 69-70
“Several studies have demonstrated that VRT is superior to CRT in terms of improving
chronic and acute athletic performance”

This sentence needs to be supported with citations.


Lines 70-74
“Specifically, several studies reported greater strength (effect size: 0.21−1.02) (Anderson et al., 2008; Arazi et al., 2020; Ataee et al., 2014; Joy et al., 2016; Katushabe & Kramer, 2020) and power (effect size: 0.35−0.49) (Joy et al., 2016; Rhea et al., 2009; Rivi Re et al., 2017) gains following VRT than CRT after a minimum of 4 weeks of training.”

There is a lot of discussion of chronic adaptations, however, your analysis based on acuate effects. The introduction should reflect that.


Lines 99-104
“To the best of our knowledge, no systematic reviews have investigated the difference in acute neuromuscular responses between VRT and CRT. Therefore, we aimed to collate evidence that directly compares the acute effects of VRT and CRT controlling for the loading schemes and the contribution of variable resistance and equipment on kinematic and kinetic variables. These findings may assist strength and conditioning practitioners in prescribing better VRT protocols.”

Please clearly state the research question or define the kinematic and kinetic variables of interest.


Lines 124-125
“(2) the study included an intervention group that used elastic bands or chains in resistance training”

Typically training refers to chronic exercise. please replace this with exercise since the nature of this study is acute responses.

Experimental design

Methods

Lines 173-176
“In the case of multiple study effects were included for studies using different contributions of
variable resistance or variable resistance equipment, the sample size of the study was divided evenly among the effects to avoid assigning disproportionate weight to these effects (Higgins & 176 Green, 2008).”

I am not sure this is the best method of addressing these issues. Please consider these methods in this paper 10.20982/tqmp.12.3.p154. this accounts for the within study variable as well as the sample size. The method currently used only addresses the weight of the sample size. Please rerun the analysis using this method.


Methods general comment.
I think more detail is needed when talking about the outcomes. How was powered measured? How was force measured on what devices? Was it the same task that was being trained? Or was it a novel task to both conditions?

Validity of the findings

Discussion
“Similar to the velocity outcomes, elastic bands result in greater effects of VRT onpeak power (p = 0.004). Based on this result, it was suggested that the effects of power adaptations from increased velocity resulting from elastic bands (Galpin et al., 2015) are much larger than those obtained from the increased force resulting from chains (Swinton et al., 2011). Thus, elastic bands may be an appropriate choice if power adaptations are desired. In addition, using the CRT higher scheme induced a greater velocity in VRT compared with CRT (Godwin et al., 2018; Heelas et al., 2021).”

A lot of the discussion seems to focus on chronic adaptions; however, you investigated the acute effects. please make sure it is clear to reader that this is speculation.

Lines 445-446
“Furthermore, higher contributions of variable resistance and using chains are advantageous for improving force”

This this study was acute claims like this cannot be make.

Additional comments

none

---

## Round 0.2 · Major Revisions

Thank you for the detailed response to the comments of the three reviewers. Although two of our reviewers are satisfied with the modifications made, a third emphasizes the need to clarify the statistical analysis, specifically the effect size, of the studies selected for the meta-analysis. For more details, please see the reviewer's comments below.

·

Basic reporting

I feel the authors have addressed my comments adequately.

Experimental design

I feel the authors have addressed my comments adequately.

Validity of the findings

I feel the authors have addressed my comments adequately.

Additional comments

The authors have addressed my comments adequately. However, I think Line 194-194 they made a mistake. The author states..."the loading at the top is equal to CRT and lower at the bottom".

This should state...the loading at the top is equal to VRT and lower at the bottom.

·

Basic reporting

The document complies with the basic aspects of the type of research proposed.

Experimental design

The design is pertinent and coherent to the problem that it is intended to address.

Validity of the findings

The statistical analysis is robust. The results manage to respond to the proposed objective and the conclusions respond to the research problem.

Additional comments

The authors have resolved most of the observations raised in the first revision.

Reviewer 3 ·

Basic reporting

no comment

Experimental design

The authors had made several substantiation improvements to the manuscript. I appreciate the authors attempt to run the analysis the way I suggested. However, I still have to main concerns. First the authors should report the analysis the when accounting the for several effect sizes coming from a single study. It does not matter that the values were close, it is important to perform the most appropriate statistical analysis. This may protect the authors form any criticism and it may serve to show others this method. My second concern is that the statistical analysis was not performed correctly. Each effect size should get its own number however this did not seem to be the case when looking at effect size id. The effect size id should be the number of effect sizes and study id should be the number of studies.

Validity of the findings

no comment

Additional comments

no comment

---

## Round 0.3 · accepted · Accept

Thanks for the detailed answer. The authors have successfully addressed all comments and concerns from reviewers.

Reviewer 3 ·

Basic reporting

no comment

Experimental design

no comment

Validity of the findings

no comment

Additional comments

no comment